# Characterization and Miniaturization of Silver-Nanoparticle Microcoil via Aerosol Jet Printing Techniques for Micromagnetic Cochlear Stimulation

**DOI:** 10.3390/s20216087

**Published:** 2020-10-26

**Authors:** Ressa Reneth Sarreal, Pamela Bhatti

**Affiliations:** College of Electrical and Computer Engineering, Georgia Institute of Technology, Atlanta, GA 30332, USA; rsarreal3@gatech.edu

**Keywords:** sensor applications, aerosol jet printing, cochlear stimulation, microcoil

## Abstract

According to the National Institute of Deafness and other Communication Disorders 2012 report, the number of cochlear implant (CI) users is steadily increasing from 324,000 CI users worldwide. The cochlea, located in the inner ear, is a snail-like structure that exhibits a tonotopic geometry where acoustic waves are filtered spatially according to frequency. Throughout the cochlea, there exist hair cells that transduce sensed acoustic waves into an electrical signal that is carried by the auditory nerve to ultimately reach the auditory cortex of the brain. A cochlear implant bridges the gap if non-functional hair cells are present. Conventional CIs directly inject an electrical current into surrounding tissue via an implanted electrode array and exploit the frequency-to-place mapping of the cochlea. However, the current is dispersed in perilymph, a conductive bodily fluid within the cochlea, causing a spread of excitation. Magnetic fields are more impervious to the effects of the cochlear environment due to the material properties of perilymph and surrounding tissue, demonstrating potential to improve precision. As an alternative to conventional CI electrodes, the development and miniaturization of microcoils intended for micromagnetic stimulation of intracochlear neural elements is described. As a step toward realizing a microcoil array sized for cochlear implantation, human-sized coils were prototyped via aerosol jet printing. The batch reproducible aerosol jet printed microcoils have a diameter of 1800 μm, trace width and trace spacing of 112.5 μm, 12 μm thickness, and inductance values of approximately 15.5 nH. Modelling results indicate that the coils have a combined depolarization–hyperpolarization region that spans 1.5 mm and produce a more restrictive spread of activation when compared with conventional CI.

## 1. Introduction

The cochlea, located in the inner ear, is a snail-like structure that exhibits a tonotopic geometry where acoustic waves are filtered spatially according to frequency (Figure 1). Throughout the cochlea, there exist hair cells that transduce sensed acoustic waves into an electrical signal that is carried by the auditory nerve to ultimately reach the auditory cortex of the brain. Thus, without functioning hair cells, hearing is imperfect or impossible. A cochlear implant bridges the gap caused by non-functional hair cells, by triggering action potentials, defined as electrical signals produced by nearby neurons, and thereby replaces a lost sensation of sound. Conventional cochlear implants (CIs) directly inject an electrical current into the surrounding tissue via implanted electrode arrays. In turn, action potentials are elicited, carried along the auditory pathway, and perceived by the brain as sound. Over 324,000 of people worldwide are CI users [1], and while many CI users achieve strong speech perception scores, there remains significant variability. One possible challenge is the spread of cochlear activation thereby diminishing the fidelity of sound. Fortunately, recent studies have shown that utilizing eddy currents generated from electromagnetic fields to stimulate nearby tissue [2], rather than direct current injection, can improve spatial resolution of the tissue in the cochlea [3] and improve CI functionality [4], and ultimately hearing perception.

However, immediate implementation of electromagnetic stimulation, micromagnetic stimulation, faces technological barriers including device size and flexibility. Namely, producing microcoils scaled for implantation in the lower chamber of the cochlea (Scala Tympani) which narrows from 1000 μm to 200 μm along the cochlear spiral. Conventional CI electrode arrays come in various forms, one shown in Figure 1E [6], and have been optimized to avoid damaging to the inner ear during implantation. While the mechanical characteristics have been improved over time, these arrays continue to be fabricated by hand which is a significant manufacturing investment.

Seeking to enable batch-fabrication, earlier work pursued cochlear electrode arrays fabricated using common bulk-substrate MEMS techniques where large amount of substrate, commonly a semi-conductor material, is etched from the desired structures [9,10]. Though MEMS techniques produce features with sub-micron resolution, resulting in appropriately sized devices for the CI application, and substrates may be flexible in nature, many of these devices demonstrate limited flexibility and cannot achieve optimal cochlear implantation [7], an example shown in Figure 1F. Avoiding insertion trauma is a primary goal of the implant; furthermore, the properties of the devices fabricated with these MEMS techniques are not suitable for this application. Thus, additive manufacturing techniques are an attractive alternative towards the ultimate goal of creating an electrode array that is safe for implantation in humans, as demonstrated by contemporary arrays, with the potential for scaled production.

Aerosol Jet Printing (AJP) is an additive manufacturing technique that is capable of producing traces of 10 μm widths and a minimum thickness of 150 nm on non-planar, flexible substrates [11]. The ability to post-process cochlear arrays with AJP microcoils is especially important as these arrays have been validated for atraumatic cochlear insertion and optimized for intracochlear placement [12]. The AJP process involves the deposition of atomized nanoparticles that are transported and focused by a sheath gas to form high-resolution structures [13,14] within the size constraints of the cochlear environment. Once validation of the coils was demonstrated from experimentation done within a Faraday cage to prevent interference, miniaturization via AJP was initiated.

## 2. Materials and Methods

This section outlines the rationale for and development process of the simulation models, inkjet-printed coils, and aerosol-jet printed coils in addition to summarizing the necessary materials for fabrication. Planar coils, rather than solenoids, have been selected for modelling as they focus greater energy at a single targeted location [15], which exhibits the potential of the planar coil for stimulation with higher spatial resolution. Importantly, magnetic fields are more impervious to the material qualities of biological environments that affect electric current and electric fields.

Our group has previously explored cochlear stimulation through finite element modeling (FEM) where the lower chamber of the cochlea was modelled to surround a model of a commercially available cylindrical, coil inductor. It was demonstrated that the coil was capable of micromagnetic stimulation of the cochlea in a feline model [3]. Moving forward from these findings, a new coil structure has been designed and tested for the cochlear application to potentially improve spatial resolution and move towards system that may reduce implantation trauma.

Briefly, FEM in COMSOL (COMSOL Multiphysics, Inc., Burlington, MA, USA) was performed to understand the behavior of the electromagnetic far-field radiation patterns generated by the coils. Then, electromagnetic characterization of neuronal stimulation was calculated to demonstrate application viability with the activating function. Next, the coils and experiments were scaled up by 100 times using inkjet printing for ease of handling during testing. As a proof-of-principle using a 3D-printed test structure, parametric experiments generating radiated field patterns were conducted to demonstrate that the fabricated coils behave similarly to the ideal scenario; namely, FEM simulations. In turn, to demonstrate proof-of-concept devices (prototypes), the process for miniaturization of the coils with AJP techniques are also detailed.

### 2.1. FEM and Electromagnetic Characterization

Figure 2 illustrates the COMSOL Multiphysics models created to observe radiation patterns of ideal silver coils, 600 μm, and the scaled, proof-of-principle 60000 μm diameter coils. FEM was primarily used as an initial testing phase demonstrating that with proper scaling, of the coil geometry and driving frequency the radiation patterns are identical, while providing a ground truth to validate fabricated coils. Thus, the cochlear environment was emulated (conductivity, permittivity, permeability) in FEM for comparison [16].

Additionally, the activating function AF(x) [17] of the microcoils was calculated to observe how nearby excitable tissue may be depolarized and hyperpolarized to characterize neural response to magnetic stimulation. Using the quasi-magnetostatics approximation of Maxwell’s equations [18], the AF of the microcoils may be found as $E+\frac{\partial \bar{A}}{\partial t}=-\nabla V$ and therefore, $AF\left(x\right)=\frac{\delta^{2}V}{\delta x^{2}}=-\nabla \left(E_{x}\left(x\right)+\frac{\partial A_{x}}{\partial t}{|}_{z}\right)$, where *E* is the electric field strength in 3D space, *V* is the electrostatic potential, $\bar{A}$ is the vector potential, and the resulting function will be in form of voltage area density.

### 2.2. Inkjet Fabrication Protocol

Inkjet printing is an additive manufacturing technique that utilizes commercially available products when compared with MEMS microfabrication techniques necessitating a cleanroom. However, inkjet printing was not selected as the first choice for the cochlear arrays. While inkjet printing is suitable for flexible surfaces such as paper and polymers, the technique is limited in two ways for the cochlear application. First, feature sizes are limited to hundreds of microns. Second, traditional cochlear implant arrays are tailored upon a silicone substrate that has been optimized over decades for atraumatic, and appropriately positioned insertion. Thus, larger-scale inkjet-printed coils were used to validate the simulation generated using FEM in COMSOL software, as described in the previous section.

The Epson Stylus C88+ Inkjet Printer (Epson America, Inc., Long Beach, CA, USA), was used to print upon a polymeric substrate of Novele IJ-200 PET-based (NovaCentrix, Austin, TX, USA). A silver ink comprised of the Metalon JS-B25P nano-silver ink by NovaCentrix with a 75 nm diameter particle size, produced the conductive traces, creating the microcoils.

The printer was prepared using NovaCentrix protocol to fully utilize conductive silver ink, rather than the intended commercial printer ink. A basic coil pattern is designed and saved as a .png or .dwg file. The coil has a 0.81 μm thickness and was allowed a 24 h resting period at 25 ${}^{\circ }$C prior to any handling for best trace continuity and substrate adhesion.

### 2.3. FEM-Injet Coil Testing

Radiation pattern comparison between the FEM model and the fabricated coils was performed to validate functionality of the fabricated coils and provide insight on radiation patterns generated from miniaturized coils. Two fabricated coils were positioned 20 cm apart, one acting as the transmitter while the other acts as the receiver, while one coil, provided a 500 mVPP, 1.6 GHz sinusoidal input voltage, was rotated at 10 degree increments as radiated field intensities were captured from the stationary receiver coil. The testing environment described in previous experimentation [16] was performed on an acrylonitrile-butadiene-styrene (ABS)-based support structure within a Faraday cage to prevent interference. Miniaturization via AJP was initiated once validation of the coils was demonstrated from this experimentation.

### 2.4. Aerosol Jet Printing Protocol

As mentioned in the introduction, AJP is an additive manufacturing technique capable of printing on planar and non-planar substrates. The Optomec Aerosol Jet 200 Series System (ANIWAA, Singapore), including the control software, KEWA, and a vial of UTDOTS AgX40 silver nanoparticles (UTDOTS, Champaign, IL, USA) were used in the creation of the coils.

A process flow diagram was designed to determine the print order of the AutoCAD coil layers which include the coil, interface pads, and an insulation bridge, all shown in Figure 3. The coil and interface pads were created via program file loaded onto the Optomec Aerosol Jet 200. The silver nanoparticle vial is exposed to an ultrasonic atomizer (UA) set to 30 ${}^{\circ }$C. The attached platen heater is set to 80 ${}^{\circ }$C. A glass slide is cleaned with isopropyl alcohol and is blow dried with nitrogen gas. After cleaning, the glass slide is secured to the platen heater. Blue painter’s tape was used to fasten the glass to the platen heater. Additional assembly is necessary to construct the nozzle with a specific tip size. A 150 μm nozzle tip is used to create the 1800 μm coils, shown in Figure 6. The sheath gas rate was set to 24 CCM and the UA gas rate is set to 12 CCM.

After the printer has been prepared, the AutoCAD file, converted into a .prg program file via VMTOOLS extension, is loaded to the control software, KEWA. Fiducials are manually entered into the program file to dictate boundaries of the printing area.

Process speed and rapid speed can be selected; 0.2 mm/s and 20 mm/s are used, respectively. Once the system is positioned to the origin/home location, the program file is initiated. As the structures are being printed, a separate hot plate is set to 200 ${}^{\circ }$C for curing the silver nanoparticles. A separate hot plate from the platen heater is used as the platen heater is limited to temperatures below 100 ${}^{\circ }$C and temperatures greater than 100 ${}^{\circ }$C are necessary to properly cure and bake the device. Once the coil and interface pads were printed, they were cured on the hot plate for 15 min. The SU-8 insulation bridge was manually applied using a thin needle and lightly brushing the appropriate location of the insulating layer, across three coil traces, then hard baked for an additional 15 min on the hot plate. The coils were then placed back onto the hot platen of the Optomec Aerosol Jet 200 to manually control the system to draw traces connecting the coil and the interface pads via KEWA control to avoid any alignment issues. The pattern was cured again at 200 ${}^{\circ }$C for 15 min on the separate hot plate.

## 3. Results

### 3.1. FEM Validation and Activating Function Analysis

The simulations conducted in COMSOL were validated with the 60,000 μm silver coils fabricated with inkjet printing techniques [14], shown in Figure 2. The fabricated coil radiation patterns demonstrated 3.3% error from the FEM radiation patterns when observing the main- beam-to-rear-beam gain ratio in addition to appropriate directivity of the fields (Figure 4).

To better understand the impact of micromagnetic stimulation on neurons, the AF was calculated for excitable tissue located a distance *z* away from the microcoil (Figure 5a). The calculation is detailed in Appendix A. The results demonstrate the spatial capability of the microcoil to stimulate excitable tissue in the cochlea. Observing AF(x) at z=300μm away from the microcoils, the approximate distance of excitable tissue from the implant, any neurons with axons arranged along the direction of the electric field will demonstrate stimulation represented by the activating function, AF(x), where η is wave impedance, κ is the wavenumber, *R* is the radius of the largest coil loop, and I(t) is the time-dependent input current. Depolarization occurs at one location along the axon, where AF(x)>0. In Figure 5, neurons oriented along the electric field, located along the x-axis where the peaks occur will be more likely to be stimulated.
(1)$$\begin{array}{cc} AF(x) & =-\frac{\eta \kappa^{3}}{4\pi }\left\{\pi \left(R^{2}+{\left(\frac{3R}{4}\right)}^{2}+{\left(\frac{2R}{4}\right)}^{2}+{\left(\frac{R}{4}\right)}^{2}\right)\right\}I\left(t\right)\\ \; & \{\left(\frac{j}{\kappa \left(x^{2}+z^{2}\right)}-\frac{1}{\kappa \sqrt{x^{2}+z^{2}}}\right)\frac{j\kappa |x|e^{-j\kappa \sqrt{x^{2}+z^{2}}}}{1000z\sqrt{x^{2}+z^{2}}\sqrt{\frac{x^{2}}{z^{2}}+1}}\\ \; & +\left(\frac{j}{\kappa \left(x^{2}+z^{2}\right)}-\frac{1}{\kappa \sqrt{x^{2}+z^{2}}}\right)\frac{\left|x\right|e^{-j\kappa \sqrt{x^{2}+z^{2}}}}{27{\left(\frac{x^{2}}{z^{2}}+1\right)}^{3/2}}\\ \; & -\left(\frac{x}{\kappa {\left(x^{2}+z^{2}\right)}^{\frac{3}{2}}}-\frac{2jx}{\kappa {\left(x^{2}+z^{2}\right)}^{2]}}\right)\frac{\left|x\right|e^{-j\kappa \sqrt{x^{2}+z^{2}}}}{1000xz\sqrt{\frac{x^{2}}{z^{2}}+1}}.\}\\ \; & +j\omega \frac{\mu_{0}I\left(t\right)R^{2}}{4}\left(e^{-j\kappa \sqrt{x^{2}+z^{2}}}\right)\\ \; & \left\{\frac{x\sqrt{(}x^{2}+z^{2}\left(-j\omega \left(x^{4}+x^{2}\left(z^{2}+R^{2}\right)\right)+2x^{4}+x^{2}\left(z^{2}-R^{2}\right)-z^{4}-R^{2}z^{2}\right)}{z^{2}\left|z\right|{\left(x^{2}+z^{2}+R^{2}\right)}^{5/2}{\left({\left(\frac{x}{z}\right)}^{2}+1\right)}^{3/2}\left|x\right|}\right\} \end{array}$$

### 3.2. AJP Coil Miniaturization and Characterization

The inkjet coils have been miniaturized by 97% using AJP techniques (Figure 6). The resulting thickness of the traces is between 11 and 14 μm, shown in Figure 7. After miniaturizing the coils, one primary challenge that persists is that the coils are fragile in the sense that if its surface is scraped by a remotely sharp object, such as a small copper wire edge, the silver can be removed, creating shorts in the coil pattern. Future application of biocompatible Parylene-C coating may resolve sensitivity to scraping as the microcoils will be fully encapsulated. Additionally, insulation layer application (SU-8 bridge) is challenging to perform manually as the center edge of the coil cannot be contacted by the insulation material; the interface pad must be connected to the center edge to fully utilize the geometry of the spiral coil and maximize radiated field patterns.

Characterization involves the application of current to the coil, then a voltage measurement across the two ends of the coil. The measurements involve a two-probe system connected to the Micromanipulator (Carson City, NV, USA) platform. Direct contact of the probes with the silver microcoil can easily damage the coil continuity which results in reduced coil length. The coils demonstrate a resistivity range of $2.33\times 10^{-7}$ to $4.54\times 10^{-7}\;\Omega \;\cdot$ m. In addition to resistivity measurements, basic inductance calculations [19] were performed to calculate the inductance of the microcoils. The coils demonstrated an inductance between 11.9 and 17.7 nH, which is a generally lower inductance and indicate that the generated fields will appear and disappear quickly as input current is applied and removed. Identical coils may be produced per printed batch if the assembly phase demonstrates no clogs or leaks in the printing system, the exact nanoparticle ink is used, and identical sheath gas rate parameters are inputted into the KEWA control program. Identical coils exhibit the same geometry and same resistivity with a tolerance of ±12%, with a range between $2.33\times 10^{-7}$ to $4.54\times 10^{-7}\;\Omega \;\cdot$ m, further outlined in Appendix B.

## 4. Discussion

### 4.1. Current Work

The current work provides a means to experimentally verify operational stimulator devices as well as a theoretical approach to characterize stimulation properties of a micromagnetic stimulator, prior to any in vitro or in vivo experimentation. COMSOL models of the microcoil present the ideal operation of the coils in the cochlea. Demonstrating fabricated coils capable of somewhat replicating results from the ideal scenario allow a level of confirmation that the fabricated coils are operational. The AF of these coils tie the modelled geometry to spatial stimulation properties in the near field region, as only the far field region was observed in COMSOL. A similar process may be performed for magnetic stimulators in other biological contexts.

The presented fabrication techniques, inkjet printing and AJP, allow for inexpensive and rapid manufacturing when compared to traditional MEMS clean room fabrication techniques. Additionally, as the utilized techniques are under the category of additive manufacturing, a minimal amount of material is necessary during development in contrast to bulk substrate manufacturing. Future steps will involve integration of the coils onto the standardized cochlear implant structures which are non-planar and are flexible. Additive manufacturing techniques, especially AJP, are capable of non-planar deposition; therefore, proceeding with AJP are likely more suitable for the cochlear implant application.

Approaching stimulation with magnetic fields rather than direct current injection from electrodes prevents a spread of excitation. Spread of excitation is caused by the conductive bodily fluids engulfing the area of implantation. Techniques used to reduce the spread of excitation include current steering; however, magnetic fields are more impervious to these effects of the cochlear environment as the material properties of bodily fluids, bones, and biological tissue exhibit relative permeability values close to 1. Typically, AF is calculated for current-injecting electrodes, but the formulation was adapted for magnetic fields. As magnetic fields pair with electric fields, the resulting electric field was used to analyze the AF (see Appendix A). The unique AF for the microcoil geometry describes the spatial functionality of the microcoils through the understanding of depolarization and hyperpolarization. Depolarization regions describe where along one axis axons in parallel with that axis will be activated, and similarly, hyperpolarization regions describe where similar axons will be deactivated. Many AFs may be calculated for various orientations; however, the calculated AF was intended to demonstrate how the tissue will be stimulated along the cochlear spiral, describing the independent channels that may be created with these microcoils.

### 4.2. Future Work

Further miniaturization will be necessary to be able to implant the microcoils into the tapering structure of the cochlea. The target size is approximately a coil with a diameter of 600 μm. A modification will be made to the fabrication protocol as the SU-8 insulation layer is currently applied by hand. The procedure will be automated with the aerosol jet printer or another insulation technique will be explored. Additionally, application of a biocompatible layer, likely made of Parylene-C, is necessary for implantation. After coating, soak tests in media modelling intracochlear fluid, likely phosphate buffer saline solution with a pH of 7.4 and a temperature of 37 ${}^{\circ }$C, will be conducted to examine stability and durability while techniques such as high-performance liquid chromatography will be performed over time to observe if any toxic contents appear in the solution. Furthermore, printing on non-planar electrode array structures and testing the array on a multi-electrode array of plated neurons is necessary to create a linear array of microcoils to stimulate restricted frequency ranges in the tonotopic, tapering geometry of the cochlea.

## 5. Conclusions

Batch-fabricated proof-of-concept 1800 μm diameter microcoils have been created. These 12 μm-thick coils demonstrate an inductance of 15.5 nH. In addition, these microcoils have achieved dimensions such that they may be implanted at the entrance of the cochlea, which has a 2 mm diameter [5]. Using AF analysis, the coils demonstrate potential to improve operating spatial resolution without exhibiting spread of excitation when compared to the conventional CIs as the coils utilize micromagnetic stimulation and the AF depolarization-hyperpolarization regions span 1.5 mm for each microcoil.

## Figures and Tables

**Figure 1 sensors-20-06087-f001:**
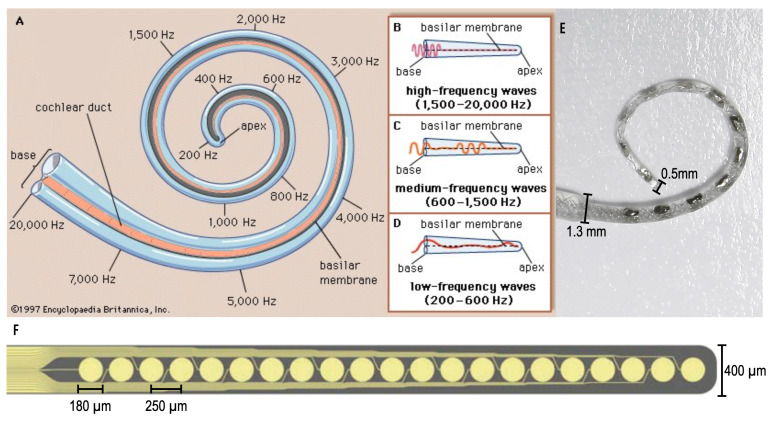
(**A**–**D**) The tonotopic geometry of the human cochlea. To a first order, acoustic frequency is mapped to place (Encyclopedia Britannica [5]). (**E**) A MED-EL cochlear implant electrode array (MED-EL Blog [6]). (**F**) A thin-film array fabricated on a polyimide substrate using commercial microelectromechanical system (MEMS) techniques [7,8].

**Figure 2 sensors-20-06087-f002:**
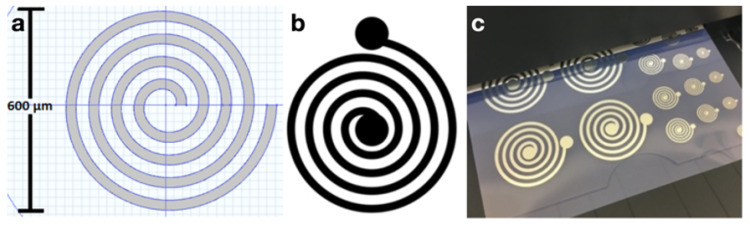
(**a**) COMSOL coil model top view. (**b**) The pattern used for the large-scale (100×) inkjet coils. (**c**) Inkjet-fabricated coils, where the largest has a diameter of 60 mm.

**Figure 3 sensors-20-06087-f003:**
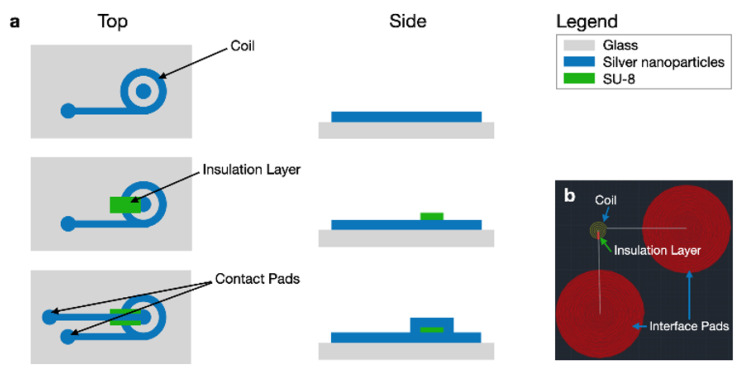
(**a**) Top and side view process flow diagrams for AJP microcoils. (**b**) The corresponding AutoCAD file.

**Figure 4 sensors-20-06087-f004:**
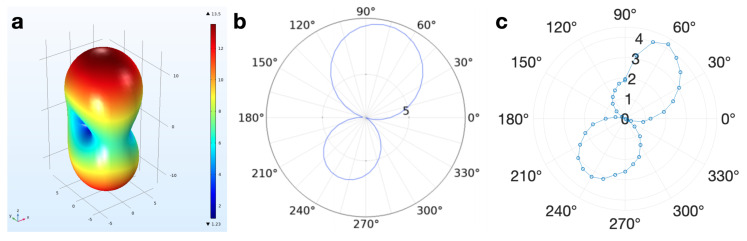
At 90 degrees, the coil pair faces are parallel with each other. (**a**) 3D depiction of the COMSOL model far-field radiated field pattern of the 60,000 μm diameter coils, which are geometrically identical to the 600 μm coils (V/m). (**b**) 2D depiction of the COMSOL model far-field radiated field pattern (V/m, 3.709 dB directivity) of the 60,000 μm diameter coils. (**c**) Average experimental far-field radiated pattern (mV across the coil length, 4.0494 dB directivity) generated from four trials.

**Figure 5 sensors-20-06087-f005:**
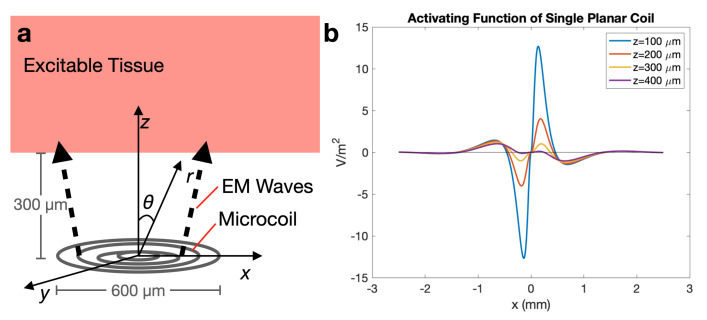
(**a**) Diagram and coordinate system for activating function; (**b**) AF(x) plot demonstrating spatially where neurons are more likely to be stimulated, when the maximum amplitude of the current is 1 A and coil distance, *z*, is varied from 100 to 400 μm. Neurons located at x=0 will likely not be stimulated by the coil.

**Figure 6 sensors-20-06087-f006:**
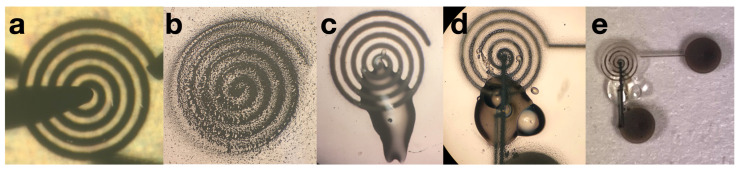
(**a**) A successfully printed AJP coil with a 12 μm thickness and 112.8 μm trace width connected to a two-probe system for characterization. (**b**) Resulting undesirable coils when printed if leaks or clogs persist during the printing process, preventable by utilizing the table in the Aerosol Jet manual. (**c**) An AJP coil with an SU-8 insulation bridge before the interface-pad printing stage. (**d**) Close up of a printed coil with insulation bridge and interface pads. The SU-8 bridge contains an air bubble, which is not desirable; however, does not impact functionality after baking and solidification. Additionally, though the interface pad trace makes contact with a portion of the coil before the end located at the center, the coil will be driven at a frequency that will resonate with the larger rings, and the magnetic fields will be negligibly affected. (**e**) The full device: coil, SU-8 insulation bridge, and interface pads.

**Figure 7 sensors-20-06087-f007:**
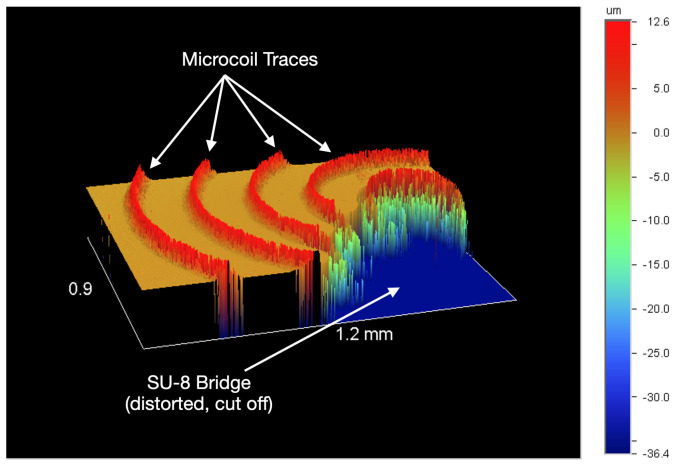
Wyko NT2000 (Veeco) profilometer partial output of a coil with an SU-8 bridge.

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
