# Peer review of "Characterization and Miniaturization of Silver-Nanoparticle Microcoil via Aerosol Jet Printing Techniques for Micromagnetic Cochlear Stimulation"

_sensors, 2020, doi:10.3390/s20216087_

Round 1
Reviewer 1 Report
The topic of paper is very interesting: it is the fabrication of microcoil by Aerosol jet printing for Coclear stimulation. It is the demonstration of the use of a technology alternative to the lithografy in application field where miniaturization and personalization are a specific requirements. The paper opens new possibilities in the field of sensors and actuators showing the possibilities to fabricate microinductance for micromagnetic effects. Other than the promising view, the paper is also scientifically sounded. Materials and method, simulation validation, requirements of the specific application, future work are correctly reported.Author Response
We thank the reviewer for the feedback and the thoughtful consideration of our work.
Reviewer 2 Report
Dear Authors,
Thank you for your contribution to the journal and science. In my opinion, the article can be accepted after major revision. Below please find the list of questions, unclearness, and suggestions:
- The first paragraph in the Introduction section should be rewritten, it is too similar to the Abstract section (it is simply not necessary to repeat all of that)
- In the Introduction, it will be sufficient to add some additional images or photos which show the current solutions (like the MEMS approach). It will improve the reader understanding
- The figures should be placed close to the text which it refers to. It will make the article more clear and easier to read (like the reference to figure 6 on page 4 and figure 6 is on page 7)
- in the Materials and Methods section, the AJP protocol has unnecessary and obvious information that is known to everyone who works with the system - I suggest delete them or move them to the appendix.
- At the end of the Results section, you have written that the tolerance of the resistance value is ±12% - please add the measured values (resistance, width, and thickness) to the appendix
- It will be more appropriate to recalculate the resistance to resistivity, for better comparison to other methods/techniques
- Why did you manually draw traces connecting the coil and the interface pads? Can you provide the figure after that process?
- Please add the measurements to the appendix (like resistance you measured) or extend the information "the coils demonstrate a resistance range of 4 to 5Ω", the same for inductance.
- If the isolation layer (SU-8) application was challenging why didn't you use the dispensing system or even the AJP?
- I suggest improving the references - for example, machine manuals are not needed
Reviewer 3 Report
The paper presents coil design for cochlear implants. Design of centimetric coils are done to check design and a novel priting technique is described to miniaturize the coils down to millimetric size.
The presented work is clearly presented but it seems to be a very preliminar result. MEMS coils have demonstrated to have far better resolutions. Besides the production cost that is mentioned, other important features that could make the inkjet printing more suitable is not discussed in the paper, such as the softness of the supporting material (paper, any polymer, PDMS ... ? instead of usual rigid as silicon or flexible as Kapton used with MEMS technologies).
COMSOL calculations and calculations of the electric field have been made with assumptions about electric and magnetic fields that should be explicited further: refs for magnetic calculations are about static or quasi static but. E and V coupling (or absence of couplig) should be briefly reminded.
FEM results can give very rich information and could/should be illustrated with more than 1 curve (figure 5)
- Ref 1 sends to a very generic page. Finding the mentioned report is really not straightforward with the given link,
- Ref 13 does not appear in IEEE Xplore
Round 2
Reviewer 2 Report
Great work, In my opinion, the article is ready to be published.